# Neuropathic-like Nociception and Spinal Cord Neuroinflammation Are Dependent on the TRPA1 Channel in Multiple Sclerosis Models in Mice

**DOI:** 10.3390/cells12111511

**Published:** 2023-05-30

**Authors:** Diéssica Padilha Dalenogare, Daniel Souza Monteiro de Araújo, Lorenzo Landini, Mustafa Titiz, Gaetano De Siena, Francesco De Logu, Pierangelo Geppetti, Romina Nassini, Gabriela Trevisan

**Affiliations:** 1Graduated Program in Pharmacology, Federal University of Santa Maria (UFSM), Santa Maria 97105-900, RS, Brazil; diessica_dalenogare@hotmail.com; 2Clinical Pharmacology Unit, Department of Health Sciences, University of Florence, 50139 Florence, Italy; daniel.souzamonteirodearaujo@unifi.it (D.S.M.d.A.); l.landini@unifi.it (L.L.); mustafa.titiz@unifi.it (M.T.); gaetano.desiena@unifi.it (G.D.S.); francesco.delogu@unifi.it (F.D.L.); pierangelo.geppetti@unifi.it (P.G.); romina.nassini@unifi.it (R.N.)

**Keywords:** microglia, astrocyte, oligodendrocyte, pain, neuroinflammation, allodynia

## Abstract

**Background**: Transient receptor potential ankyrin 1 (TRPA1) activation is implicated in neuropathic pain-like symptoms. However, whether TRPA1 is solely implicated in pain-signaling or contributes to neuroinflammation in multiple sclerosis (MS) is unknown. Here, we evaluated the TRPA1 role in neuroinflammation underlying pain-like symptoms using two different models of MS. **Methods**: Using a myelin antigen, *Trpa1^+^*^/*+*^ or *Trpa1^−^*^/*−*^ female mice developed relapsing-remitting experimental autoimmune encephalomyelitis (RR-EAE) (Quil A as adjuvant) or progressive experimental autoimmune encephalomyelitis (PMS)-EAE (complete Freund’s adjuvant). The locomotor performance, clinical scores, mechanical/cold allodynia, and neuroinflammatory MS markers were evaluated. **Results**: Mechanical and cold allodynia detected in RR-EAE, or PMS-EAE *Trpa1^+^*^/*+*^ mice, were not observed in *Trpa1^−^*^/*−*^ mice. The increased number of cells labeled for ionized calcium-binding adapter molecule 1 (Iba1) or glial fibrillary acidic protein (GFAP), two neuroinflammatory markers in the spinal cord observed in both RR-EAE or PMS-EAE *Trpa1^+^*^/*+*^ mice, was reduced in *Trpa1^−^*^/*−*^ mice. By Olig2 marker and luxol fast blue staining, prevention of the demyelinating process in *Trpa1^−^*^/*−*^ induced mice was also detected. **Conclusions**: Present results indicate that the proalgesic role of TRPA1 in EAE mouse models is primarily mediated by its ability to promote spinal neuroinflammation and further strengthen the channel inhibition to treat neuropathic pain in MS.

## 1. Introduction

Multiple sclerosis (MS), an autoimmune disease associated with extensive inflammatory and demyelinating processes in the brain and spinal cord [1], is classified into three subtypes: relapsing–remitting multiple sclerosis (RRMS), primary progressive multiple sclerosis (PPMS), and secondary progressive multiple sclerosis (SPMS) [2]. RRMS (85% of the diagnoses) is characterized by neurological dysfunction and partial recovery periods [3]. MS subtypes are characterized by intense neurodegenerative processes, resulting in a progressive worsening of physiological functions. Mouse models that recapitulate RRMS and PMS encompass experimental autoimmune encephalomyelitis (EAE) evoked by Quil A adjuvant (RR-EAE) or complete Freund’s adjuvant (CFA, PMS-EAE) in animals treated with antigen myelin oligodendrocyte glycoprotein_35-55_ (MOG_35-55_) [4,5]. In this autoimmune disease, neuropathic pain is a common sensory debilitating symptom, but the underlying mechanisms remain poorly understood. In particular, the implication of the neuroinflammatory process in MS pain symptoms continues not to be completely investigated.

The transient receptor potential ankyrin 1 (TRPA1) is a cation channel activated by irritant compounds, including the prototypic compound allyl isothiocyanate, and by a large series of reactive endogenous agonists, such as hydrogen peroxide and 4-hydroxynonenal [6,7,8]. The original identification of TRPA1 in a heterogeneous subpopulation of primary sensory neurons, mostly co-localized with TRP vanilloid 1 (TRPV1), has driven research on the channel role to signal pain from the peripheral nervous system to the brain [9].

TRPA1 is expressed in central and peripheral glial cells [10,11], such as astrocytes [12] oligodendrocytes [13] and Schwann cells [14], being associated with ischemia-related neural damage [13] and allodynia [15]. The activation of inflammatory cells in the central nervous system has been documented in RR-EAE mice [16]. Enhancement of cells expressing glial fibrillary acidic protein (GFAP+ve), a specific marker of activated astrocytes, and an increase in ionized calcium-binding adapter molecule 1 (Iba1+ve) cells, a marker of macrophages/microglia, has been detected in the corpus callosum, hippocampus, and lumbar spinal cord section from RR-EAE mice [16]. A possible role of TRPA1 has been reported in models of MS, and in a mouse model of the demyelinating disease induced by cuprizone, where the activation of astrocyte expressing TRPA1 elicits the release of pro-inflammatory mediators contributing to oligodendrocyte apoptosis [17]. We recently observed that the increase in mechanical and cold allodynia in mouse RR-EAE and PMS-EAE models was attenuated after the TRPA1 pharmacological blockade [4,5], providing the involvement of the channel in the mechanical/cold hypersensitivity in both mice models of MS. However, the implications of the blockade of the central or peripheral TRPA1 and its ability to promote neuroinflammation in EAE models remain unknown. 

Here, by using TRPA1 wild-type or genetically deleted female mice, we first confirm that TRPA1 is essential for the development of mechanical and cold allodynia in both RR-EAE and PMS-EAE. Second, we show that TRPA1 genetic deletion attenuated the increased expression of inflammatory markers, including Iba1^+^ and GFAP^+^ in the spinal cord, and the Olig2 marker and luxol fast blue staining, markers for demyelinating process, in both RR-EAE and PMS-EAE mouse models. 

## 2. Materials and Methods

### 2.1. Animals

The following mouse strains were used: littermate wild-type (*Trpa1**^+/+^*), and TRPA1-deficient (*Trpa1^−^*^/*−*^) [18] (female, 15–30 g). All the animals (five per cage) were accommodated with wood-shavings bedding and nesting material and were maintained in controlled temperatures (22 ± 2 °C) and bred in-house with a 12-h light/dark cycle (lights on 7:00 am to 7:00 pm). Water and laboratory standard animal food (Charles River, Milan, Italy) were provided ad libitum. According to the ethical guidelines, experiments were performed to investigate pain in conscious animals [19], and the Italian Ministry of Health (protocol #1194/2015-PR) approved the experimental procedures. Behavioral studies followed the Animal Research Reporting In Vivo Experiments (ARRIVE) guidelines [20]. Animals were moved and acclimatized to the experiment room for at least one hour before each procedure. All experiments were performed by an operator blinded to drug administration and genotype.

### 2.2. MS Models Induction

To induce the model of PMS-EAE, an emulsion containing 200 μg of MOG_35-55_ antigen was dissolved in phosphate-buffered saline (PBS) in an equal volume of CFA supplemented with 400 μg of *Mycobacterium tuberculosis.* H37Ra extract [4,21] was administered by subcutaneous (s.c.) injection in the flank region of the mouse. In the non-immunized mice (control), MOG_35-55_ was not added to the mixture. 

The mouse model of RR-EAE was caused by an injection (s.c.) of a mixed solution of MOG_35–55_ antigen (200 μg) and Quil A (45 μg) in PBS solution (100 μL) [5,16,22] in the flank region of the mouse. Control mice received only equal doses of Quil A without MOG_35-55_ [5,16,22]. All mice received 300 ng (PMS-EAE) or 250 ng (RR-EAE) of pertussis toxin intraperitoneally (i.p.), on the day of induction and 48 h later [4,21].

### 2.3. Assessment of EAE Clinical Signs 

The clinical signs of the PMS-EAE model were measured using a clinical scale that evaluated the neurological impairment using scores [21]. Mice were assessed using this scale: grade 0, regular mouse; grade 1, flaccid tail (disease onset); grade 2, mild hind limb weakness with quick righting reflex; grade 3; severe hind limb weakness with slow righting reflex; and grade 4, hind limb paralysis in one hind limb or both. Mice were monitored on different days post-induction (p.i.; 3–14 days) of the PMS-EAE model to assess the clinical signs [4,23,24].

The clinical disease scoring paradigm for RR-EAE-induced mice was assessed weekly during the experimental period p.i. (7–35 days) according to the following scale: 0, normal behavior; 0.5, limpness of the distal tail region and hunched appearance; 1, utterly limp tail or developing weakness in the hind limbs; 1.5, limp tail and distinct hind limbs weakness recognized by unsteady gait and poor grip of hind limbs while hanging on cage underside; 2, limp tail with unilateral partial hind limb paralysis; 2.5, limp tail and partial paralysis of bilateral hind limbs; 3, complete paralysis of bilateral hind limbs; 3.5, complete bilateral hind limbs paralysis and unilateral fore limb paralysis; and 4, quadriplegia. In the RR-EAE model, clinical scores ≤0.5 indicated no disease or disease remission [5,16,25]. 

As we evaluated two different MS models, the days of clinical score evaluation differed. Moreover, the PMS-EAE is a more aggressive model of the disease that mimics a progressive subtype of the disease in humans, as previously optimized [21]. Similarly, the RR-EAE mimics the relapsing–remitting MS in humans, a different form of the disease requiring more time to present the characteristic signs of the disease form, also previously reported [16].

As exclusion criteria previously described, animals displaying a clinical grade >1.5 (RR-EAE) or ≥2 (PMS-EAE) would be removed from the study [4,5,23]. The nociceptive tests require the mice not to present an elevated clinical score to assess the evoked stimulus area with von Frey filaments and acetone tests. Mice were also monitored after PMS- or RR-EAE induction to assess body weight. If an animal showed a weight loss of 20–30% of the initial weight, the animal was excluded from the experimental setting. However, as reported in our previous research, none of the animals was excluded from the study [4,5].

### 2.4. Locomotor Activity Testing 

The locomotor activity was observed on different days before and after PMS- or RR-EAE induction. First, mice were trained on the rotarod apparatus (47650 Rota-Rod NG, Ugo Basile, Gemonio (VA), Italy) one day before induction. Mice were placed on a spinning cylinder for 60 s, at a fixed speed of 16 rpm, and the latency to fall from the apparatus was recorded. The rotarod test was performed on days 3, 5, 7, 9, 11, 13, and 14 (PMS-EAE) or 7, 14, 21, 28, and 35 (RR-EAE) after EAE induction [21,23,26]. Animals that failed to remain for 180 s on the rotarod were removed from the study [4,5,23]. 

### 2.5. Mechanical and Cold Allodynia Evaluation

The development of mechanical allodynia was evaluated by placing mice in individually transparent boxes on a wire mesh platform allowing easy access to the right hind paw plantar surface. Filaments of different stiffness were applied to the plantar surface of the hind paw, ranging from 0.07 to 2.0 g (0.07, 0.16, 0.40, 0.60, 1.0, 1.4, 2.0 g). According to the up-and-down paradigm, the mechanical threshold was obtained [27,28]. The mechanical paw-withdrawal threshold (in g) response was calculated from the resulting scores [29]. The animals were acclimatized for 60 min to determine the mechanical thresholds and the test, and all animals were assessed before PMS or RR-EAE induction (baseline values). The mechanical threshold was evaluated on days 3, 5, 7, 9, 11, 13, and 14 for PMS-EAE or 7, 14, 21, 28, and 35 for RR-EAE p.i and in control animals.

The cold allodynia was assessed by the acetone drop test. Acetone (20 μL) was applied to the plantar surface of the right hind paw, and the time spent lifting, licking, or wagging the paw was counted for 60 s [30,31,32]. Cold allodynia was evaluated before PMS- or RR-EAE induction and on days 3, 5, 7, 9, 11, 13, and 14 for the PMS-EAE or 7, 14, 21, 28, and 35 for the RR-EAE p.i. and in control animals. The nociceptive tests, mechanical threshold, and acetone tests were performed on the same set of days and in the same set of animals as in our previous studies [4,5].

### 2.6. Immunofluorescence

Spinal cord (L4-L6) samples were obtained, on the 14th or 35th day after induction from the *Trpa1^+^*^/*+*^ e *Trpa1^−^*^/*−*^ PMS- or RR-EAE-induced mice, respectively. To collect the tissues, animals were transcardially perfused with PBS, followed by 4% paraformaldehyde. Spinal cord samples were removed, post-fixed for 24 h, and cryoprotected (4 °C, overnight) in 30% sucrose. Cryosections (40 µm) of the spinal cord were incubated (4 °C, overnight) with the following primary antibodies: Iba1 [1:1000, Wako, catalog #019–19741, rabbit monoclonal, Wako, Osaka, Japan], GFAP [1:500, Z0334, rabbit monoclonal, DAKO, Santa Clara, CA, USA], anti-Olig2 (1:100, MABN50, Millipore, Darmstadt, DE) diluted in PBS and 2.5% normal goat serum (NGS). Sections were then incubated with fluorescent secondary antibodies: polyclonal Alexa Fluor^®^ 488, and polyclonal Alexa Fluor^®^ 594 (1:600, Invitrogen, Waltham, MA, USA) (2 h, room temperature), and cover slipped. Analysis of negative controls (nonimmune serum) was simultaneously performed to exclude the presence of non-specific immunofluorescent staining, cross-immunostaining, or fluorescence bleed-through. Tissues were visualized, and digital images were captured using a Zeiss Axio Imager 2 microscope with Z-stacks in the Apotome mode (Carl Zeiss Microscopy GmbH, Jena, Germany). Data are expressed as mean fluorescence intensity (% of basal).

### 2.7. Luxol Fast Blue Staining 

Spinal cord slices were placed in a solution of 1:1 alcohol/chloroform for 2 h and hydrated on 95% ethyl alcohol. After that, they were left on luxol fast blue solution at 56 °C for 2 h. The differentiation process was made by putting slices in a lithium carbonate solution for 30 s, 70% ethyl alcohol for 30 s, and washing with distilled water. This differentiation process was repeated 5–7 times until we achieved the best staining. The analysis was made using Image-J (NIH, Bethesda, MD, USA) to measure the stained area. 

### 2.8. Statistical Analysis

Data were expressed as mean + SEM and analyzed statistically by the parametric one-way or two-way analysis of variance according to the experimental protocol, followed by the post-test Bonferroni when needed. The individual values were inserted as column statistics in Prism GraphPad (GraphPad Prism 8.0 program, Boston, MA, USA) and the mean of these values was calculated. To meet parametric assumptions, data of mechanical threshold scores were log-transformed before analyses for the mechanical allodynia test. To calculate the number of cells/areas, a ZEN 2.6 blue edition (Carl Zeiss Microscopy GmbH, Jena, Germany) program was used. Differences between groups were considered significant when *p* values were less than 0.05 (*p* < 0.05) (GraphPad Prism 8.0 program).

## 3. Results

### 3.1. TRPA1 Genetic Deletion Prevents the Development of Mechanical and Cold Allodynia in the RR-EAE Mouse Model

RR-EAE *Trpa1^+^*^/*+*^ mice developed a time-dependent increase in the mechanical and cold allodynia (Figure 1A,B) from day 21 to 35 p.i. Mice with genetic deletion of TRPA1 did not develop either mechanical or cold allodynia after RR-EAE induction compared to *Trpa1^+^*^/*+*^ mice (Figure 1A,B). Thus, TRPA1 deletion prevents the development of neuropathic pain-like symptoms. The mouse body weight did not present a significant difference (Figure 1C) during the days of the experiment for all the groups. Furthermore, all experimental groups did not present significant locomotor activity changes assessed in the rotarod test (Figure 1D).

### 3.2. TRPA1 Genetic Deletion Prevents the Development of Mechanical and Cold Allodynia after PMS-EAE Induction

Similarly, in the PMS-EAE model, the induced *Trpa1^+/+^* mice showed the development of mechanical and cold allodynia (Figure 2), which was absent in PMS-EAE *Trpa1^−/−^* mice (Figure 2A,B). Moreover, the body weight for both strains did not present a significant difference between PMS-EAE and control groups (Figure 2C). All the animals did not present locomotor activity changes assessed in the rotarod test (Figure 2D).

### 3.3. Iba1 Marker Is Reduced in the Spinal Cord of Trpa1^−/−^ Mice Induced to RR- or PMS-EAE Models

We also explored the expression of Iba1, a marker for activated microglia, in the spinal cord, in the two MS mouse models. Iba1+ve cells were increased in *Trpa1^+/+^* mice in both RR- and PMS-EAE-induced mice (Figure 3). The increase in the number of Iba1+ve cells was observed either in the total area of the spinal cord (Figure 3B,C) or in the dorsal horn area (Figure 3E,F) for both EAE mouse models. A strong reduction in the number of Iba1+ve cells in the spinal cord slices from RR- or PMS-EAE *Trpa1^−/−^* was observed in the total area of the spinal cord (Figure 3B,C) and in the dorsal horn area (Figure 3E,F).

### 3.4. GFAP Marker Is Reduced in the Spinal Cord of Mice Trpa1^−/−^ Induced to RR- or PMS-EAE Models

We also investigated the astrocyte activation in the two MS models. We observed an increase in GFAP+ve cells in *Trpa1^+/+^* mice after the RR-EAE which was significantly reduced in *Trpa1^−/−^* mice, in the total spinal cord area and dorsal horn area (Figure 4A–C) after the RR-EAE induction. Similar results were observed in mice induced to PMS-EAE (Figure 4D–F). The increase in GFAP+ve cells numbers was observed in the total area and dorsal horn of the spinal cord from *Trpa1^+/+^* induced mice and it was reduced in *Trpa1^−/−^* PMS-EAE induced mice (Figure 4D–F).

### 3.5. The Demyelinating Process, but Not the Development of Clinical Signs, Is Reduced in RR- and PMS-EAE Trpa1^−/−^ Mice

We also explored whether TRPA1 deletion was able to reduce the development of clinical signs and demyelination in both RR- and PMS-EAE models. We observed that the development of clinical signs for RR- or PMS-EAE (Figure 5A,B) was the same for *Trpa1^+/+^* or *Trpa1^−/−^*, indicating the absence of prevention by TRPA1 deletion. However, the luxol fast blue staining (myelin marker) showed the reduction in myelin in the spinal cord from *Trpa1^+/+^* RR- (Figure 5C–E) and PMS-EAE (Figure 5D–F) mice, which was not observed in the spinal cord from *Trpa1^−/−^* RR- and PMS-EAE mice. 

### 3.6. TRPA1-Deleted Mice Showed Prevention of Demyelinating Process after RR- or PMS-EAE Induction

The demyelinating process was also observed in the decrease in the oligodendrocyte proliferation marker, Olig2, in *Trpa1^+/+^* RR- (Figure 6A–C) and PMS-EAE (Figure 6D–F) induced mice. Surprisingly, the TRPA1 deletion reduced the demyelinating process observed by restoring the number of Olig2^+^ve cells in the total area (Figure 5B) and dorsal horn (Figure 5C) of the spinal cord of RR-EAE induced mice. A similar prevention was observed in the number of Olig2^+^ve cells of PMS-EAE-induced mice in the total area (Figure 5E) and dorsal horn (Figure 5F) of the spinal cord.

## 4. Discussion

The role of the TRPA1 channel in the development of pain symptoms has been extensively investigated [33]. In addition, the EAE induction model is the most common MS rodent model used in the research field due to its clinical and pathological similarities to human disease, which include neuroinflammation, demyelination, and neuron loss [34]. Previously, we showed that the administration of TRPA1 antagonists transiently reduced the mechanical and cold allodynia developed after the RR- or PMS-EAE inductions [4,5]. Here, first, we confirmed the involvement of TRPA1 in the mechanical and cold allodynia in the two different models of MS, the RR- and PMS-EAE models, by using TRPA1-deleted mice. Second, we investigated the involvement of TRPA1 in spinal cord neuroinflammation after the induction of these two EAE mouse models. 

TRPA1 genetic deletion prevents the development of mechanical and cold allodynia either in the RR- or PMS-EAE mouse model. Although the TRPA1 deletion did not prevent the development of clinical scores, it prevented the demyelinating process after RR- or PMS-EAE induction. Furthermore, TRPA1 deletion significantly reduced the levels of neuroinflammatory markers, Iba1 and GFAP, in spinal cord tissues which were increased in *Trpa1^+/+^* following PMS- or RR-EAE. Using a genetic approach via the total TRPA1 deletion, we contend that these findings suggest an essential role of TRPA1 in neuropathic pain-like symptoms and neuroinflammation in the two different models of MS.

Neuropathic pain is one of the most prevalent pain symptoms in MS patients [35,36]. Different mouse models of neuropathic pain showed that TRPA1 deletion reduced the mechanical allodynia in the paw and periorbital area [14,15,32,37,38]. In the alcoholic neuropathy and partial sciatic nerve ligation model, the TRPA1 total deletion or its specific silencing in Schwann cells prevented the development of mechanical and cold allodynia [14,37]. In addition, TRPA1-deleted mice did not develop mechanical and cold allodynia in other models of neuropathic pain, including chemotherapy-induced peripheral neuropathy [38] and complex regional pain syndrome [15] Furthermore, the development of periorbital mechanical allodynia by the constriction of the infraorbital nerve was also attenuated following TRPA1 deletion [32]. Here, in agreement with all the previous neuropathic pain models, the total deletion of the TRPA1 channel was also able to prevent the development of cold and mechanical allodynia after RR- or PMS-EAE induction. These results corroborate our previous results obtained following the TRPA1 pharmacologic blockade (by two selective TRPA1 antagonists, HC-030031 and A967079) in the PMS and RR-EAE mouse models [4,5].

TRPA1 receptors are expressed in the central nervous system on astrocytes [12,39,40] and oligodendrocytes [13]. The activation of astrocytes, microglia, and oligodendrocytes has emerged as a leading mechanism underlying chronic pain [41]. Several studies showed the upregulation of glial markers such as Iba1 and GFAP [41] and cell morphological alterations, including hypertrophy and/or proliferation. Accordingly, three different studies using the EAE mouse model showed an upregulation of Iba1 markers in induced mice [16,42,43]. In particular, the increased expression of the Iba1+ve microglia marker in the corpus callosum indicates a prominent role of these cells in the neuroinflammatory process induced by the PMS-EAE model [42]. In the same induction protocol for RR-EAE, the increase in the Iba1 marker in the spinal cord was also shown [16]. Other evidence established a relation between TRPA1 activation, the Iba1 marker, and nociception behavior [44]. In a previous study, the TRPA1 deletion prevented the microglia activation, by modulating the Iba1, in a pain model induced by the bradykinin B1 receptor agonist, thus revealing an important role for this channel in this type of nociceptive behavior [44]. Furthermore, some neuropathic models, such as chronic constriction injury of the sciatic nerve [45,46] and chemotherapy-induced peripheral neuropathy [47], displayed the development of mechanical allodynia in induced mice as well as an increase in the iba1 marker in the spinal cord. Our previous results obtained by real-time PCR showed the enhancement of Iba1 mRNA levels in the RR-EAE model [5]. Here, we indicate that the TRPA1 deletion could prevent an increase in the Iba1+ve cells caused by RR- and PMS-EAE induction in the spinal cord. 

Astrocytes perform diverse functions in the central nervous system, and some evidence showed their essential role in neurodegenerative diseases, such as MS [48,49]. Other findings showed the importance of the astrocytes in MS, which influences the clinical phenotypes, pathogenesis, progression, and recovery of this pathology [48]. The TRPA1 is also expressed in astrocytes, and recent studies in an Alzheimer’s mouse model reported that its activation by oligomeric forms of amyloid-β peptide promotes astrocytic Ca^2+^ hyperactivity and synaptic dysfunction [50], and by increasing GFAP levels a neuroinflammatory process [51]. Different findings, using the EAE mouse models, showedthat the upregulation of GFAP [16,52] correlates with MS physiopathology. The increase in GFAP expression was also related to pain-like behaviors in a model of neuropathic pain induced by spinal cord ligation [53,54,55]. Here, we reported that the deletion of TRPA1 prevents the increase in GFAP+ve cells observed in the wild-type mice induced in the two different MS mouse models. 

Recently, TRP channels have been related to the neurological diseases that identified their roles in calcium ion homeostasis [56]. The regulation of Ca^2+^ is involved in physiological pathways such as exocytosis at synapses, neuronal survival, proliferation, differentiation, and gene transcription. However, in a neurological disease, the Ca^2+^ homeostasis was disrupted and is related to mitochondrial dysfunction, increased reactive oxygen species (ROS), and mitochondrial dysfunction [56]. Interestingly, several studies have reported a correlation between intracellular Ca^2+^ concentrations and other pathogenic mechanisms, including an imbalance between the antioxidant function and ROS production [57] and mitochondrial dysfunction [58]. Furthermore, a specific mechanism involving TRPA1 and Alzheimer’s disease (AD) was demonstrated using APP/PS1 Tg/TRPA1 KO mice. The TRPA1 knockout mice presented exacerbation in spatial learning and memory deficits, increased Ab deposition, the release of pro-inflammatory cytokines, and inhibition of transcriptional factor NF-kB activities, as well as a nuclear factor of activated T cells [51]. Thus, future studies could investigate the role of these different mechanisms that may cause TRPA1 activation.

The reactive astrogliosis presents different protein markers beyond the GFAP marker, such as the cluster of differentiation receptors (CD), connexins (Cx), the sphingosine-1 phosphate receptor 3, synemin, the complement component 3, and metallothionein [59]. In addition, some other microgliosis markers such as histone-lysine N-methyltransferase SETD7 are involved in the development of spinal microgliosis and neuropathic pain induction [60]. Further studies are required to better investigate the downstream markers for active astrogliosis and microgliosis. Despite the limitations of our results and considered analysis, other genetic and morphological parameters of astrogliosis and microgliosis may be investigated in future studies in multiple sclerosis models. Nevertheless, our results complement our previous publications [4,5], whereas, in the literature, a comparison of the MS mouse models and all the approaches used in these present results have not been described until now.

The demyelinating process and oligodendrocyte damage in MS results in the impairment of neuron impulse propagation and, consequently, in several functional deficits [61]. As previously described, the EAE models induce the demyelinating process, which manifests motor impairments mimicking MS clinical manifestations [34,62]. Here, we observed the presence of clinical signs after PMS- or RR-EAE-induced mice. However, the deletion of TRPA1 channels did not prevent the development of these clinical signs. Similar results were also observed in another EAE model, where the blocking of toll-like receptors 2 and 4 could not reduce the motor impairments but reversed the allodynia caused by EAE induction [63].

The Olig2 marker is highly expressed in motor neurons and oligodendrocytes and leads to the differentiation and maturation of oligodendrocyte progenitor cells [64]. This transcript factor regulates myelin regeneration during chronic demyelination and represents an important marker for investigating a possible new target to restore myelination [64]. In the cuprizone model, the demyelination was related to a reduction in luxol fast blue staining and a significant reduction in the cell numbers marked by Olig2 [65]. Here, we reported a reduction in Olig2 and luxol fast blue staining in the spinal cord in PMS- and RR-EAE-induced mice, and TRPA1 deletion showed a prevention of the demyelinating process. Overall, our results suggest that TRPA1 channel blockade may improve the remyelinating process, although it is not sufficient for recovering from clinical signs induced by the MS models. 

Several cell types and proteins are involved in the inflammatory and degenerative processes related to the development of neurodegenerative diseases [56]. Recently, the importance of the receptor–ligand interaction demonstrated a potential field to develop a specific treatment for the different outcomes of multiple sclerosis [66]. A recent study showed the main role of some ligand–receptor pairs in active and chronic lesion type; other specific pairs were related to active and remyelinating lesions, and the interaction of cytokine ligands and chemokine receptors related to the active lesion in MS. All these results demonstrated potential targets to treat the different outcomes of MS [66]. Thus, for future investigations, analyzing the receptor–ligand interaction may clarify TRP receptors’ involvement and their link with MS. Additionally, different receptors have been investigated for MS treatment, such as sigma-1 ligands that are involved in excitotoxicity, calcium dysregulation, mitochondrial and endoplasmic reticulum dysfunction, inflammation, and astrogliosis. This ligand may activate the neurotrophic factors and neural plasticity similar to disease-modifying agents used in the treatment of central nervous system (CNS) diseases [67]. 

Thus, previous results and our current findings suggest a possible role for TRPA1 in the inflammatory process and attenuation of neuropathic pain-like behaviors in two different mouse models of MS [4,5,68,69]. In conclusion, the reported findings strongly support the theory that the TRPA1 channel is a valuable target for future investigations to treat neuropathic pain in MS patients. 

## Figures and Tables

**Figure 1 cells-12-01511-f001:**
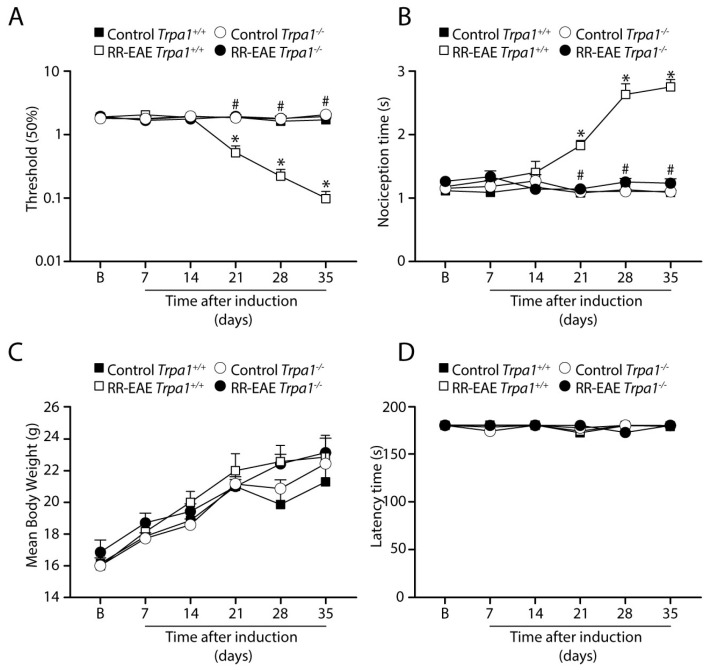
**TRPA1 genetic deletion prevents the development of mechanical and cold allodynia in relapsing–remitting experimental autoimmune encephalomyelitis (RR–EAE) mouse model.** Evaluation of (**A**) mechanical allodynia, (**B**) cold allodynia, (**C**) body weight, and (**D**) latency to fall in the rotarod test. Data are mean + SEM (*n* = 6) for (**A**–**D**). (**B**) for baseline valued before RR-EAE induction. * *p* < 0.05, when compared to Control *Trpa1^+^*^/*+*^ vs. RR-EAE *Trpa1^+^*^/*+*^; and ^#^ *p* < 0.05 when compared to PMS-EAE *Trpa1^+/+^* vs. RR-EAE *Trpa1^−^*^/*−*^ [Two-way ANOVA, followed by Bonferroni’s post hoc test].

**Figure 2 cells-12-01511-f002:**
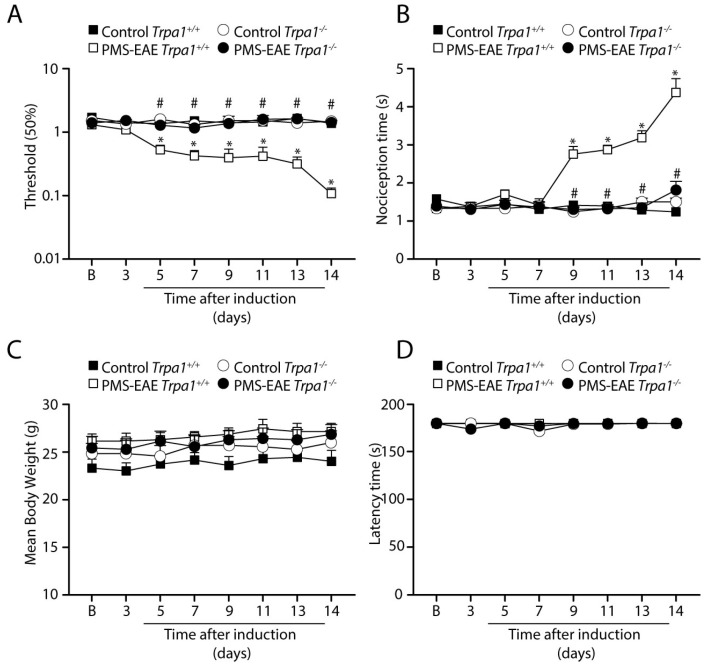
**TRPA1 genetic deletion prevents the development of mechanical and cold allodynia in the progressive experimental autoimmune encephalomyelitis (PMS–EAE) mouse model.** Evaluation of (**A**) mechanical allodynia, (**B**) cold allodynia, (**C**) body weight, and (**D**) latency to fall in the rotarod test. Data are mean + SEM (*n* = 6) for (**A**–**D**). (**B**) for baseline valued before PMS-EAE induction. * *p* < 0.05, when compared to Control *Trpa1^+/+^* vs. progressive PMS-EAE *Trpa1^+/+^*; and ^#^ *p* < 0.05 when compared to PMS-EAE *Trpa1^+/+^* vs. PMS-EAE *Trpa1^−/−^* [Two-way ANOVA, followed by Bonferroni’s post hoc test].

**Figure 3 cells-12-01511-f003:**
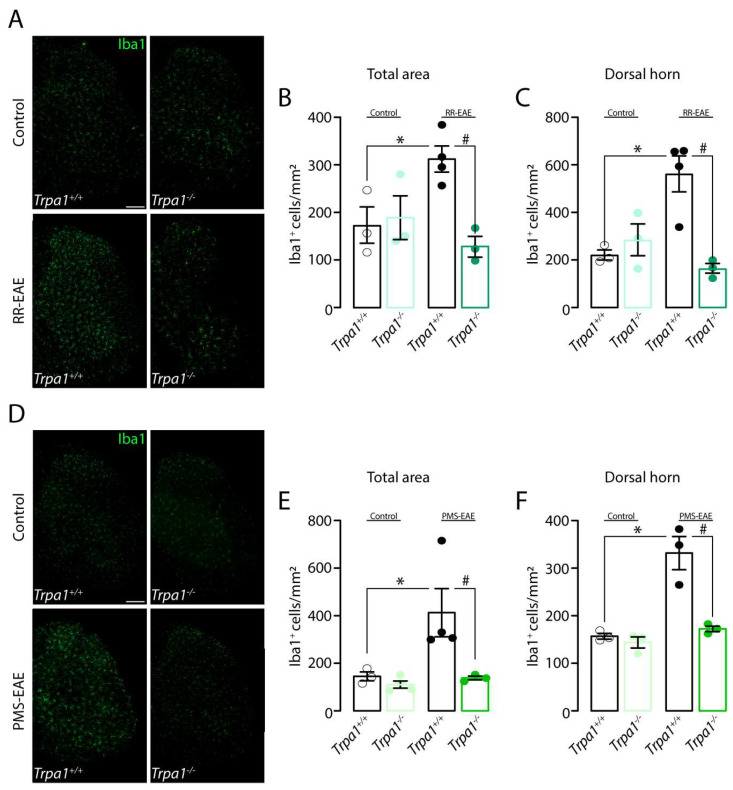
**Iba1 (ionized calcium-binding adapter molecule 1) marker level was reduced in the spinal cord of *Trpa1^−/−^* mice induced to relapsing–remitting (RR–EAE) or progressive experimental autoimmune encephalomyelitis (PMS–EAE).** (**A**) Representative photomicrographs of spinal cord area for *Trpa1^+/+^* or *Trpa1^−/−^* after 35 days of the RR-EAE induction, cumulative data of Iba1+ve cells in (**B**) total spinal cord, and (**C**) dorsal horn. (**D**) Representative photomicrographs of spinal cord area for *Trpa1^+/+^* or *Trpa1^−/−^* after 14 days of the PMS-EAE induction, cumulative data of Iba1+ve cells in the (**E**) total spinal cord, and (**F**) dorsal horn. Data are mean + SEM (*n* = 4) for Graphs (**A**–**F**). * *p* < 0.05, when compared to Control *Trpa1^+/+^* vs. RR-EAE or PMS-EAE *Trpa1^+/+^*; and ^#^ *p* < 0.05 when compared to RR-EAE or PMS-EAE *Trpa1^+/+^* g vs. PMS-EAE *Trpa1^−/−^* [One-way ANOVA, followed by Bonferroni’s post hoc test].

**Figure 4 cells-12-01511-f004:**
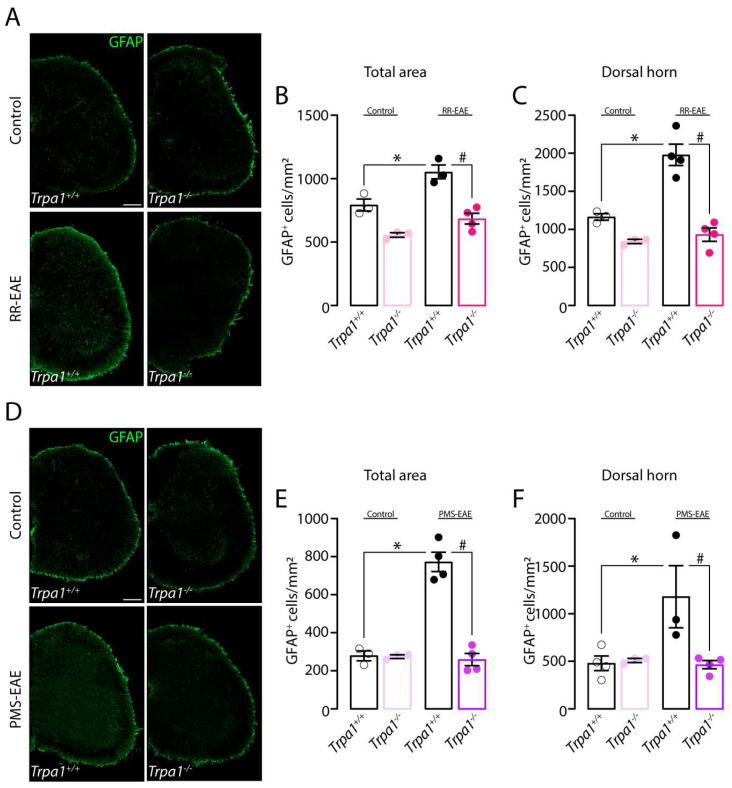
**The astrocyte marker (GFAP, glial fibrillary acidic protein) was reduced in the spinal cord of *Trpa1^−/−^* mice induced to relapsing–remitting (RR–EAE) or progressive experimental autoimmune encephalomyelitis (PMS–EAE).** (**A**) Representative photomicrographs of spinal cord area for *Trpa1^+/+^* or *Trpa1^−/−^* after RR-EAE, (**B**) cumulative data of GFAP+ve cells in the total spinal cord, and (**C**) dorsal horn. (**D**) Representative photomicrographs of spinal cord area for *Trpa1^+/+^* or *Trpa1^−/−^* after PMS-EAE, (**E**) GFAP+ve cells in the spinal cord, and (**F**) dorsal horn. Data are mean + SEM (*n* = 4) for Graphs (**A**–**F**). * *p* < 0.05, when compared to Control *Trpa1^+/+^* vs. RR-EAE or PMS-EAE *Trpa1^+/+^*; and ^#^ *p* < 0.05 when compared to RR-EAE or PMS-EAE *Trpa1^+/+^* vs. PMS-EAE *Trpa1^−/−^* [One-way ANOVA, followed by Bonferroni’s post hoc test].

**Figure 5 cells-12-01511-f005:**
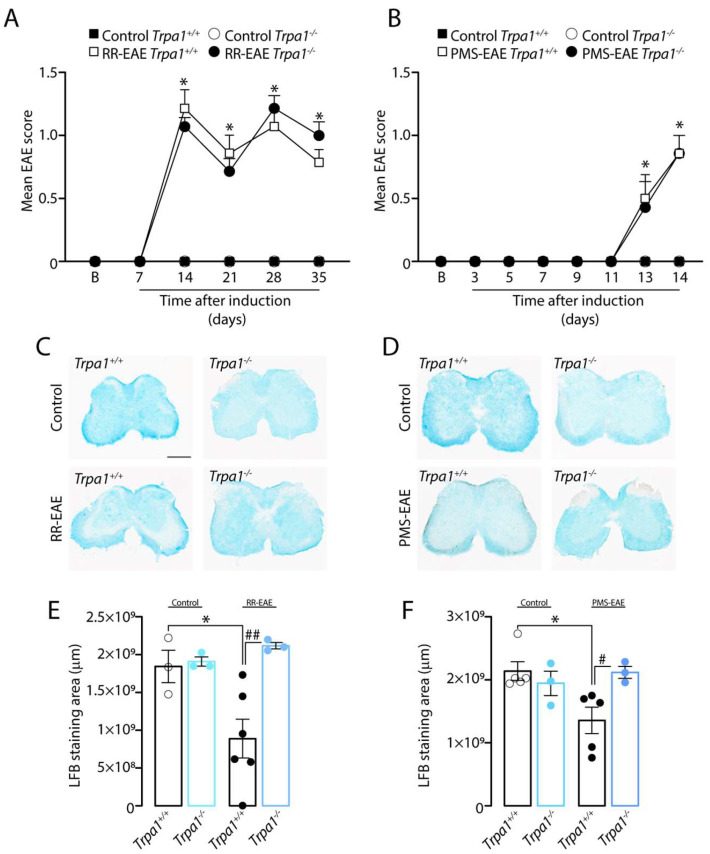
**TRPA1 deletion prevents the demyelinating process, but not clinical signs, after relapsing–remitting (RR–EAE) or progressive experimental autoimmune encephalomyelitis (PMS–EAE) induction.** (**A**) Clinical scores in RR-EAE mice; (**B**) Clinical scores in PMS-EAE mice; (**C**) representative photomicrographs of luxol fast blue staining for RR-EAE; (**D**) representative photomicrographs of luxol fast blue staining for PMS-EAE; (**E**) Luxol fast blue staining for RR-EAE; and (**F**) Luxol fast blue staining for PMS-EAE. Data are mean for Graphs (**A**,**B**) and + SEM (*n* = 6) and Graphs (**E**,**F**) (*n* = 4). * *p* < 0.05, when compared to Control *Trpa1^+/+^* vs. RR-EAE or PMS-EAE *Trpa1^+/+^*; and ^#^ *p* < 0.05, ^##^
*p* < 0.01, when compared to RR-EAE or PMS-EAE *Trpa1^+/+^* vs. PMS-EAE *Trpa1^−/−^* [One-way ANOVA, followed by Bonferroni’s post hoc test].

**Figure 6 cells-12-01511-f006:**
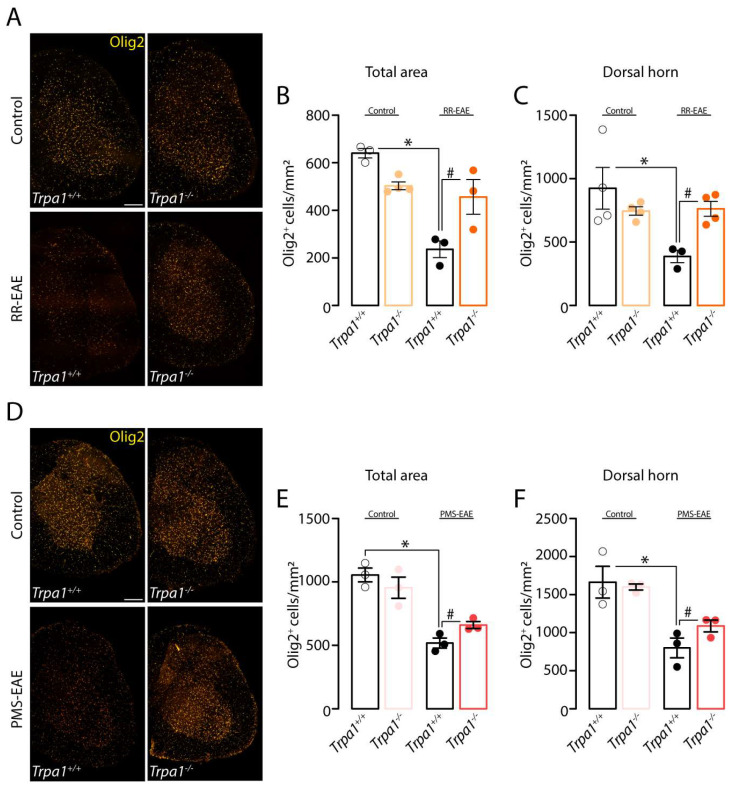
**TRPA1 knockout mice showed prevention of demyelinating process after relapsing–remitting (RR–EAE) or progressive experimental autoimmune encephalomyelitis (PMS–EAE) induction.** (**A**) Representative photomicrographs of Olig2 in the spinal cord area for *Trpa1^+/+^* or *Trpa1^−/−^* after RR-EAE, cumulative data of Olig2^+^ cells in (**B**) total spinal cord, (**C**) dorsal horn. (**D**) Representative photomicrographs of the spinal cord for *Trpa1^+/+^* or *Trpa1^−/−^* after PMS-EAE, cumulative data of Olig2^+^ cells in (**E**) the total spinal cord, and (**F**) dorsal horn. Data are mean + SEM (*n* = 4) for Graphs (**A**–**F**). * *p* < 0.05, when compared to Control *Trpa1^+/+^* vs. RR-EAE or PMS-EAE *Trpa1^+/+^*; and ^#^ *p* < 0.05 when compared to RR-EAE or PMS-EAE *Trpa1^+/+^* vs. PMS-EAE *Trpa1^−/−^* [One-way ANOVA, followed by Bonferroni’s post hoc test].

## Data Availability

All data generated or analyzed during this study are included in this article. Further inquiries can be directed to the corresponding author.

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
