# Peer review of "Neuropathic-like Nociception and Spinal Cord Neuroinflammation Are Dependent on the TRPA1 Channel in Multiple Sclerosis Models in Mice"

_cells, 2023, doi:10.3390/cells12111511_

Round 1
Reviewer 1 Report
Dalenogare et al. demonstrated that TRPA1 genetic deletion prevents the development of neuropathic pain, reactive gliosis, and demyelination in two different EAE models. The findings seem simple and concrete. However, TRPA1's critical role in neuropathic pain has already been well documented since Kwan et al. published a paper titled "TRPA1 Contributes to Cold, Mechanical, and Chemical Nociception but Is Not Essential for Hair-Cell Transduction" in Neuron in 2006. Furthermore, a previous report by the same group has already suggested the critical role of TRPA1 in neuropathic pain (Dalenogare et al., 2020, Experimental Neurology). These previous reports could diminish the impact of the current report.
The clear protection from reactive astrogliosis and microgliosis in TRPA1-KO mice seems interesting, but GFAP-positive and Iba1-positive cell counting may not be sufficient. This reviewer recommends that an in-depth analysis of astrogliosis and microgliosis would significantly increase the impact of this study.
Author Response
We would like to thank you very much for the important and constructive suggestions given for our manuscript. They will certainly improve its scientific value.
REVIEWER ‘1’: “Dalenogare et al. demonstrated that TRPA1 genetic deletion prevents the development of neuropathic pain, reactive gliosis, and demyelination in two different EAE models. The findings seem simple and concrete. However, TRPA1's critical role in neuropathic pain has already been well documented since Kwan et al. published a paper titled "TRPA1 Contributes to Cold, Mechanical, and Chemical Nociception but Is Not Essential for Hair-Cell Transduction" in Neuron in 2006. Furthermore, a previous report by the same group has already suggested the critical role of TRPA1 in neuropathic pain (Dalenogare et al., 2020, Experimental Neurology). These previous reports could diminish the impact of the current report.”
Point 1: The clear protection from reactive astrogliosis and microgliosis in TRPA1-KO mice seems interesting, but GFAP-positive and Iba1-positive cell counting may not be sufficient. This reviewer recommends that an in-depth analysis of astrogliosis and microgliosis would significantly increase the impact of this study.”
Response: We have inserted this limitation in the revised version of our manuscript lines 408-418: “The reactive astrogliosis present different protein markers beyond of GFAP marker, such as the cluster of differentiation receptor (CD), connexins (Cx), sphingosine-1 phosphate receptor 3, synemin, complement component 3, and metallothionein (doi: 10.3390/biom11091361). In addition, some other microgliosis markers as histone-lysine N-methyltransferase SETD7 are involved in the development of spinal microgliosis and neuropathic pain induction (doi: 10.1016/j.bbi.2019.09.007). Further studies are needed to better investigate the downstream markers for active astrogliosis and microgliosis. Despite the limitation of our results and analysis considered, other genetic and morphological parameters of astrogliosis and microgliosis might be investigated in future studies in multiple sclerosis models. Nevertheless, our results are complementing our previous publications (10.1016/j.expneurol.2020.113241; 10.1007/s12035-020-01891-9), whereas, in the literature, the comparison of the MS mouse models, and all the approaches used in these present results was not described until now.”
We hope that answer to your request has been satisfactorily improved and is now suitable for publication in Cells - MDPI.
Sincerely yours,
Professor Gabriela Trevisan

Reviewer 2 Report
Major revision
Interesting paper looking at Trpa1 effects for MS.
Figure 2 is valuable in that it looks at how deletion influences allodynia.
The paper would benefit from expansion into looking at the ligand receptors PMID: 37008452.
Additionally, paper would benefit from discussion of sigma receptors PMID: 28315269.
If the above are addressed and references included, paper would be of interest.
Author Response
First, we would like to thank you very much for the important and constructive suggestions given for our manuscript. They will certainly improve its scientific value.
Point 1: “Interesting paper looking at Trpa1 effects for MS. Figure 2 is valuable in that it looks at how deletion influences allodynia. The paper would benefit from expansion into looking at the ligand receptors PMID: 37008452. Additionally, paper would benefit from discussion of sigma receptors PMID: 28315269. If the above are addressed and references included, paper would be of interest.”
Response: We included the suggested references in lines 440-452 of the revised manuscript: "Recently, the importance of receptor-ligand interaction demonstrated a potential field to develop a specific treatment for the different outcomes of multiple sclerosis (doi: 10.51737/2766-4503.2022.044). A recent study showed the main role of some ligand-receptor pairs in active and chronic lesion types, other specific pairs were related to active and remyelinating lesions, and the interaction of cytokine ligands and chemokine receptors related to the active lesion in MS. All these results demonstrated potential targets to treat the different outcomes of MS (doi: 10.51737/2766-4503.2022.044). Thus, for future investigations, analyzing the receptor-ligand interaction might clarify TRP receptors' involvement and its link with MS.
Also, different receptors had been investigated for MS treatment, such as sigma-1 ligands that are involved in excitotoxicity, calcium dysregulation, mitochondrial and endoplasmic reticulum dysfunction, inflammation, and astrogliosis. This ligand may activate the neurotrophic factors and neural plasticity similar to disease-modifying agents used in the treatment of central nervous system (CNS) diseases (doi: 10.1007/978-3-319-50174-1_10)"
We hope that the new version of our manuscript has been satisfactorily improved and is now suitable for publication in Cells – MDPI.
Sincerely yours,
Professor Gabriela Trevisan

Reviewer 3 Report
Neuropathic-like nociception and spinal cord neuroinflammation are dependent on the TRPA1 channel in multiple sclerosis 3 models in mice.
The present study explains the TRPA1 receptor in neuropathic pain-like symptoms. Two types of animals, Trpa1+/+ and Trpa1-/- with two different study models a relapsing-remitting experimental autoimmune encephalomyelitis (RR-EAE) (Quil A as adjuvant) or progressive multiple sclerosis (PMS)-EAE (complete Freund’s adjuvant) were used to study the impact of TRPA1 receptor in pain. The study has its merits and the results are more valuable.
There are some minor issues found in the manuscript:
1. A huge number of typographical errors are found in the entire manuscript. Need a careful review.
2. There is an issue with the body weight of the mice used in RR-EAE models. In line 82, it is mentioned that 20-30 g of the body weight. When refer the figure 1C, the body weight is not matched.
3. Lines 112 and 115, why does the period differ between PMS-EAE and RR-EAE? Explain in the manuscript.
4. Line 123, “Animals displaying a clinical grade> 1.5 (RR-EAE) or ≥ 2 (PMS-EAE), were .removed from the study”. It would be good to add an explanation here.
5. Lines 147-153, two sets of experiments, mechanical threshold, and acetone test were performed in the same set of days. There were conducted with the same set of animals or different ones?
Author Response
Manuscript # cells2305796
RESPONSE TO THE COMMENTS OF REVIEWER 3’
First, we would like to thank you very much for the important and constructive suggestions given for our manuscript. They will certainly improve its scientific value.
REVIEWER 3’: “Neuropathic-like nociception and spinal cord neuroinflammation are dependent on the TRPA1 channel in multiple sclerosis 3 models in mice. The present study explains the TRPA1 receptor in neuropathic pain-like symptoms. Two types of animals, Trpa1+/+ and Trpa1-/- with two different study models a relapsing-remitting experimental autoimmune encephalomyelitis (RR-EAE) (Quil A as adjuvant) or progressive multiple sclerosis (PMS)-EAE (complete Freund’s adjuvant) were used to study the impact of TRPA1 receptor in pain. The study has its merits, and the results are more valuable. There are some minor issues found in the manuscript”:
Point 1: “A huge number of typographical errors are found in the entire manuscript. Need a careful review.”
Response: As requested, we revised all the text.
Point 2: “There is an issue with the body weight of the mice used in RR-EAE models. In line 83, it is mentioned that 20-30 g of the body weight. When refer the figure 1C, the body weight is not matched.”
Response: As requested, we corrected the body weight description in line 83 as follows in red: “(female, 15-30 g)”.
Point 3: “Lines 112 and 115, why does the period differ between PMS-EAE and RR-EAE? Explain in the manuscript.”
Response: As requested, we added the following paragraph in the section "2.3 Assessment of EAE clinical signs" in lines 124-129 in red: As we evaluated two different MS models, the days of clinical score evaluation differ. Moreover, the PMS-EAE is a more aggressive model of the disease that mimics a progressive subtype of the disease in humans, as previously optimized (doi: 10.1016/j.pain.2008.11.002). Similarly, the RR-EAE mimics the relapsing-remitting MS in humans, a different form of the disease requiring more time to present the characteristic signs of the disease form, also previously reported 10.1016/j.pbb.2014.09.003).".
Point 4: “Line 123, “Animals displaying a clinical grade> 1.5 (RR-EAE) or ≥ 2 (PMS-EAE), were removed from the study”. It would be good to add an explanation here.”
Response: As requested, we rewrote the paragraph in lines 130-136 in red: As exclusion criteria previously described, animals displaying a clinical grade > 1.5 (RR-EAE) or ≥ 2 (PMS-EAE), would be removed from the study (10.1016/j.expneurol.2012.12.012; 10.1016/j.expneurol.2020.113241; 10.1007/s12035-020-01891-9). The nociceptive tests require the mice not to present an elevated clinical score to assess the evoked stimulus area with von Frey filaments and acetone test. Mice were also monitored after PMS- or RR-EAE induction to assess body weight. If an animal showed a weight loss of 20-30% of the initial weight, the animal was excluded from the experimental setting. However, as reported in our previous research none of the animals were excluded from the study (10.1016/j.expneurol.2020.113241; 10.1007/s12035-020-01891-9).
Point 5: “Lines 147-153, two sets of experiments, mechanical threshold, and acetone test were performed in the same set of days. There were conducted with the same set of animals or different ones?”
Response: The nociceptive tests, mechanical threshold, and acetone tests were performed in the same set of days and in the same set of animals as in our previous studies (10.1016/j.expneurol 2020.113241; 10.1007/s12035-020-01891-9). We described this information in lines 162-164.
We hope that the new version of the manuscript has been satisfactorily improved and is now suitable for publication in Cells - MDPI.
Sincerely yours,
Professor Gabriela Trevisan

Round 2
Reviewer 1 Report
I regret to say that the authors did not properly address this reviewer's comment. First, the authors did not address the comment about novelty. To improve a novelty of this study, this reviewer had suggested an in-depth analysis of astrogliosis and microgliosis, which is apparently doable with already obtained confocal images, but the authors did not perform.
Reviewer 2 Report
Accept
Reviewer 3 Report
The authors have revised the manuscript and addressed all the clarifications.